# The export receptor Crm1 forms a dimer to promote nuclear export of HIV RNA

David S Booth[1], Yifan Cheng[2]*, Alan D Frankel[2]*

[1]Graduate Group in Biophysics, University of California, San Francisco, San Francisco, United States; [2]Department of Biochemistry and Biophysics, University of California, San Francisco, San Francisco, United States

**Abstract** The HIV Rev protein routes viral RNAs containing the Rev Response Element (RRE) through the Crm1 nuclear export pathway to the cytoplasm where viral proteins are expressed and genomic RNA is delivered to assembling virions. The RRE assembles a Rev oligomer that displays nuclear export sequences (NESs) for recognition by the Crm1-Ran$^{GTP}$ nuclear receptor complex. Here we provide the first view of an assembled HIV-host nuclear export complex using single-particle electron microscopy. Unexpectedly, Crm1 forms a dimer with an extensive interface that enhances association with Rev-RRE and poises NES binding sites to interact with a Rev oligomer. The interface between Crm1 monomers explains differences between Crm1 orthologs that alter nuclear export and determine cellular tropism for viral replication. The arrangement of the export complex identifies a novel binding surface to possibly target an HIV inhibitor and may point to a broader role for Crm1 dimerization in regulating host gene expression.

## Introduction

Human immunodeficiency virus (HIV) replication depends on the coordinated expression of viral proteins from fully spliced, partially spliced, and unspliced forms of its RNA genome. Only fully spliced RNAs encoding viral regulatory proteins initially reach the cytoplasm for translation (*Feinberg et al., 1986*). Later in the life cycle, the regulatory protein Rev directs the nuclear export of unspliced and partially spliced viral RNAs by engaging the Crm1 nuclear export pathway typically used to transport host protein cargoes and small RNAs (*Sodroski et al., 1986*; *Emerman et al., 1989*; *Felber et al., 1989*; *Kohler and Hurt, 2007*; *McCloskey et al., 2012*).

Rev coopts Crm1 to export viral RNAs by bridging the interaction between Crm1 and the Rev Response Element (RRE), a structured RNA located within an intron of unspliced and singly-spliced viral RNAs (*Figure 1A*). The amino terminal portion of Rev recognizes the RRE through an arginine rich motif flanked by hydrophobic domains that facilitate Rev oligomerization on the RRE (*Malim and Cullen, 1991*). The Rev nuclear export sequence (NES), located near the carboxy terminus, recruits Crm1 in cooperation with Ran bound to GTP (Ran$^{GTP}$) to elicit nuclear export of viral RNAs (*Fischer et al., 1995*).

A set of structures detail the architecture of the RRE (*Fang et al., 2013*), Rev binding to a single sites in the RRE (*Battiste et al., 1996*), Rev oligomerization (*Daugherty et al., 2010b*; *DiMattia et al., 2010*), and the Crm1-NES interaction (*Güttler et al., 2010*), but the organization of the entire export complex has remained unknown. Without a complete structure, it is unclear, for example, how HIV modulates Rev-RRE activity through mutations in the RRE that change the RNA fold yet minimally affect Rev binding (*Legiewicz et al., 2008*; *Sloan et al., 2013*) or how subtle amino acid substitutions between nearly identical orthologs of Crm1 restrict HIV replication in different species (*Nagai-Fukataki et al., 2011*; *Sherer et al., 2011*; *Elinav et al., 2012*). Here we report the overall architecture of the Rev-RRE/Crm1-Ran$^{GTP}$ export complex using single-particle electron microscopy (EM).

*For correspondence: ycheng@ucsf.edu (YC); frankel@cgl.ucsf.edu (ADF)

**Competing interests:** The authors declare that no competing interests exist.

**eLife digest** To be able to multiply, viruses first have to infect a host cell and then hijack the host's molecular machinery to make viral proteins. The first stage of this process takes place in the nucleus of the host cell and involves the DNA being transcribed to make RNA molecules. These RNA molecules must then be exported from the nucleus to the cytoplasm, where the proteins are made.

For RNA molecules that have been transcribed from the cell's own DNA, this export process happens automatically. However, the export of viral RNA molecules requires help from the virus. In the case of HIV-1, the virus supplies a protein called Rev, which binds to a region on the viral RNA molecules called the Rev Response Element. The Rev protein then binds to a group of host proteins called the Crm1 export complex to send the viral RNA molecules to the cytoplasm. Although the 3D structures of the individual components have been worked out, it is not known how the viral RNA molecule, the Rev protein and the Crm1 proteins all fit together to make a complex.

Booth et al. have used a technique called single-particle electron microscopy to produce a 3D structure of the whole complex. It shows that this complex forms with two Crm1 proteins contacting each other as they bind to the Rev protein that is already bound to the RNA molecule. It also reveals a new surface of the complex that had not been previously predicted to exist. In parallel work from the same laboratory, *Jayaraman et al., 2014*. have used a different technique to reveal a highly-detailed 3D structure of Rev molecules binding to the Rev Response Element.

Both structures shed new light on how the HIV-1 virus is able to multiply in its host, which may aid future efforts to develop new treatments for the disease.

Unexpectedly, Rev-RRE forms a complex with an ordered Crm1 dimer, unlike previously characterized cargoes that bind a Crm1 monomer (*Dong et al., 2009*). Residues unique to simian primates form a dimerization interface between Crm1 monomers that is seen as a crystal contact in the structure of human Crm1 (*Dong et al., 2009*). It appears that RRE-directed remodeling of the Rev oligomer (*Jayaraman et al., 2014*) may organize the Rev NESs to enhance avidity for the Crm1 dimer and help tune its export activity. It will be interesting to determine if some cellular protein-RNA complexes may also utilize Crm1 dimerization to tune nuclear export.

## Results

### Two copies of Crm1 and Ran$^{GTP}$ bind the Rev-RRE complex

In order to define the arrangement of HIV RNA export complexes for structural studies, we first determined the biochemical requirements for complex formation using recombinant Crm1, Ran, Rev tagged at the amino-terminus with a GB1 domain (GB1-Rev), and a 245 nt portion of RRE RNA transcribed in vitro. Crm1 co-purified with GB1-Rev in the presence of both Ran$^{GTP}$ and the RRE (*Figure 1B*, lane 11), but showed only a weak interaction with Ran$^{GDP}$ or without the RRE (compare lane 11 to lanes 12 and 9). The interaction between Crm1 and Rev requires a functional NES, as GB1-Rev bearing a dominant-negative M10 mutation in the NES, Leu78Asp and Glu79Leu (*Malim et al., 1989*), did not bind Crm1 under any condition (lanes 17–18 and 21–24). These results mirror in vivo and in vitro experiments showing that Rev uses an intact NES to bridge the interaction between the RRE and Crm1 utilizing Ran$^{GTP}$ (*Fischer et al., 1995*; *Fornerod et al., 1997*; *Askjaer et al., 1998*). Furthermore, our data directly show for the first time that the RRE enhances the Rev-Crm1 interaction, supporting the hypothesis that the Rev NES must be recognized in the proper Rev-RRE context to function (*Askjaer et al., 1999*).

Previously, we demonstrated that the RRE assembles six subunits of Rev into discrete, asymmetric complexes with a total mass of 160 kD (*Daugherty et al., 2010a*). We used size-exclusion chromatography to further determine if these Rev-RRE complexes can associate with Crm1 and Ran$^{Q69L}$, a mutant that stabilizes the GTP-bound form of Ran (*Bischoff et al., 1994*). When increasing amounts of Crm1-Ran$^{Q69L}$ were added to a fixed concentration of Rev-RRE, all four components eluted in a 462 kD fraction (*Figure 1C,D*), corresponding to two copies of Crm1 and Ran$^{Q69L}$ per Rev-RRE complex (*Figure 1—figure supplement 1*). In fact, negatively stained electron micrographs of complexes from this fraction

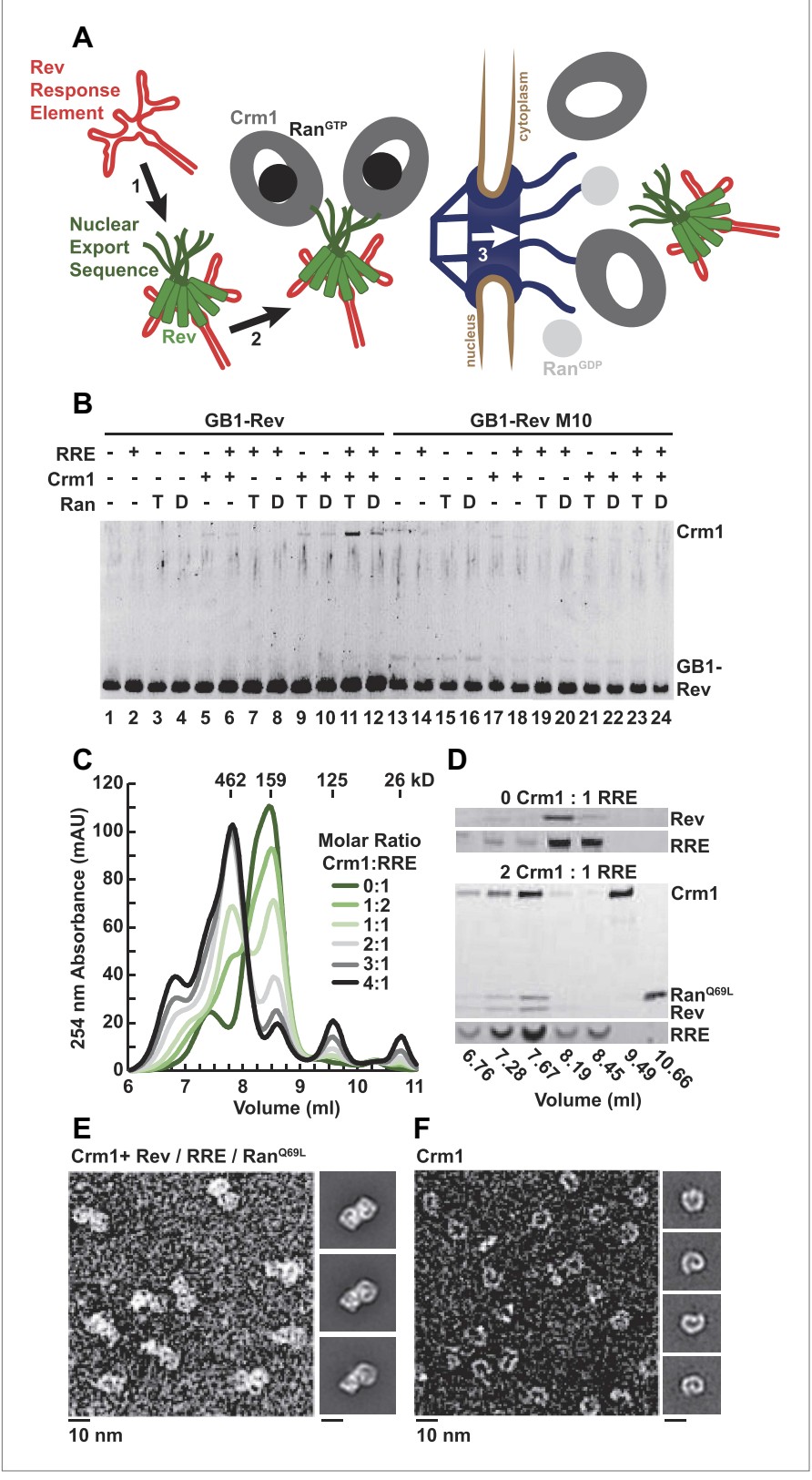

**Figure 1**. A dimer of Crm1-Ran[GTP] binds the Rev-RRE complex. (**A**) Rev assembles on the Rev Response Element (RRE) (step 1) and presents nuclear export sequences (NESs) for Crm1 to recognize in conjunction with Ran[GTP] (step 2). The assembled particle passes through the nuclear pore complex (step 3) to the cytoplasm where Ran GTP

*Figure 1. Continued on next page*

*Figure 1. Continued*

hydrolysis stimulates the dissociation of Rev-RRE from Crm1. (**B**) The gel shows the protein composition from affinity purifications of GB1-Rev incubated in the presence (+) or absence (−) of the RRE, Crm1, and Ran. T or D denotes Ran exchanged with GTPγS or GDP, respectively. RevM10 is a mutation within the NES of Leu78Asp and Glu79Leu (*Malim et al., 1989*). The gel is representative of three independent experiments. (**C–D**) Reconstituted Rev-RRE particles preferentially bind two copies of Crm1 and Ran$^{GTP}$. (**C**) Size-exclusion chromatograms of Rev-RRE with increasing molar ratios of Crm1 and Ran$^{Q69L}$ show saturation at a molar ratio of two Crm1 per RRE. Excess Crm1 eluted as free monomers (9.49 ml fraction) or in a small population of larger complexes (6.76 and 7.02 ml fractions). Molecular masses were determined using multi-angle laser light scattering after size-exclusion chromatography and have a technical error of three percent. (**D**) The composition of peak fractions was analyzed by SDS-PAGE for proteins and by Urea-PAGE for RNA. These results are representative of two independent experiments. (**E–F**) Negative-stain electron micrographs show two copies of Crm1 in complexes reconstituted with Rev-RRE and Ran$^{GTP}$. Representative fields and class averages of 7196 nuclear export particles (**E**) and 6445 Crm1 molecules alone (**F**) are shown.

The following figure supplement is available for figure 1:

**Figure supplement 1**. Assessment of the stoichiometry of Crm1, Ran$^{GTP}$, Rev, and the RRE.

---

clearly show two Crm1 molecules per single particle, compared to images of Crm1 monomers alone (*Figure 1E,F*). A Crm1 dimer explains the need for at least two NESs in the Rev oligomer to function (*Hoffmann et al., 2012*), and it may also be important for efficiently transporting the RRE through the nuclear pore complex since the number of receptors required for nuclear transport scales with the size of the cargo (*Ribbeck and Görlich, 2002*). A Crm1 monomer is thought to use a protein adaptor to export relatively small, structured RNAs (*McCloskey et al., 2012*). In contrast, the RRE is almost twice the size of host RNA cargoes, is embedded in a much larger mRNA, and is bound to the Rev oligomer.

## Species-specific residues define an interface between Crm1 monomers

Previous structures of Crm1 bound to host cargo proteins have shown monomeric binding, so we sought to understand how Rev-RRE organizes a Crm1 dimer. We utilized single-particle negative-stain EM and a random conical tilt method (*Radermacher, 1988*) to generate a 3D reconstruction of the Crm1-Ran$^{GTP}$/Rev-RRE complex (*Figure 2—figure supplement 1*). We recorded two images of the same sample area, one tilted to 60° and one at 0°, and assigned particles from the 0° images into classes. We then calculated 3D reconstructions for each class using corresponding particle images from the 60° tilted specimen. All reconstructions produced the same structure but in different orientations, permitting us to align and then average the classes to generate a final structure of the complex at ~25 Å resolution.

In this reconstruction, we observed two connected bowl-shaped densities, reminiscent of the known Crm1 morphology (*Dolker et al., 2013*), arranged with twofold symmetry. To place Crm1 into the reconstruction, we initially fit the crystal structure of two murine Crm1-Ran$^{GTP}$ monomers (*Güttler et al., 2010*) into the EM density (*Figure 2—figure supplement 2*), even though our complex was assembled with the human receptor, because Ran$^{GTP}$ is absent in the human Crm1 crystal structure (*Dong et al., 2009*) and is known to bind on the interior of Crm1 and stabilize a strained conformation that opens the hydrophobic NES binding cleft (*Monecke et al., 2009, 2013*). Remarkably, seven residues specific to simian primates that lie outside the NES-binding site and are known to be important for Rev function and HIV biogenesis (*Nagai-Fukataki et al., 2011*; *Sherer et al., 2011*; *Elinav et al., 2012*) were symmetrically apposed in this arrangement of the Crm1 dimer. This fitting suggested that the species-specific residues may form a protein–protein interface between the Crm1 monomers, and indeed, we discovered that this same interface formed a crystal contact between human Crm1 monomers in the crystal structure bound to Snurportin1 (*Dong et al., 2009*). We further found that a Crm1 dimer held together by this symmetric interface unambiguously fit as a rigid body into the EM map (*Figure 2A*).

This Crm1 dimer interface was not previously noted because Crm1 was known to bind the host Snurportin1 cargo as a monomer and was seen as a monomer in the asymmetric unit of the crystal. Subsequent studies interpreted species-specific residues in the context of the Crm1 monomer

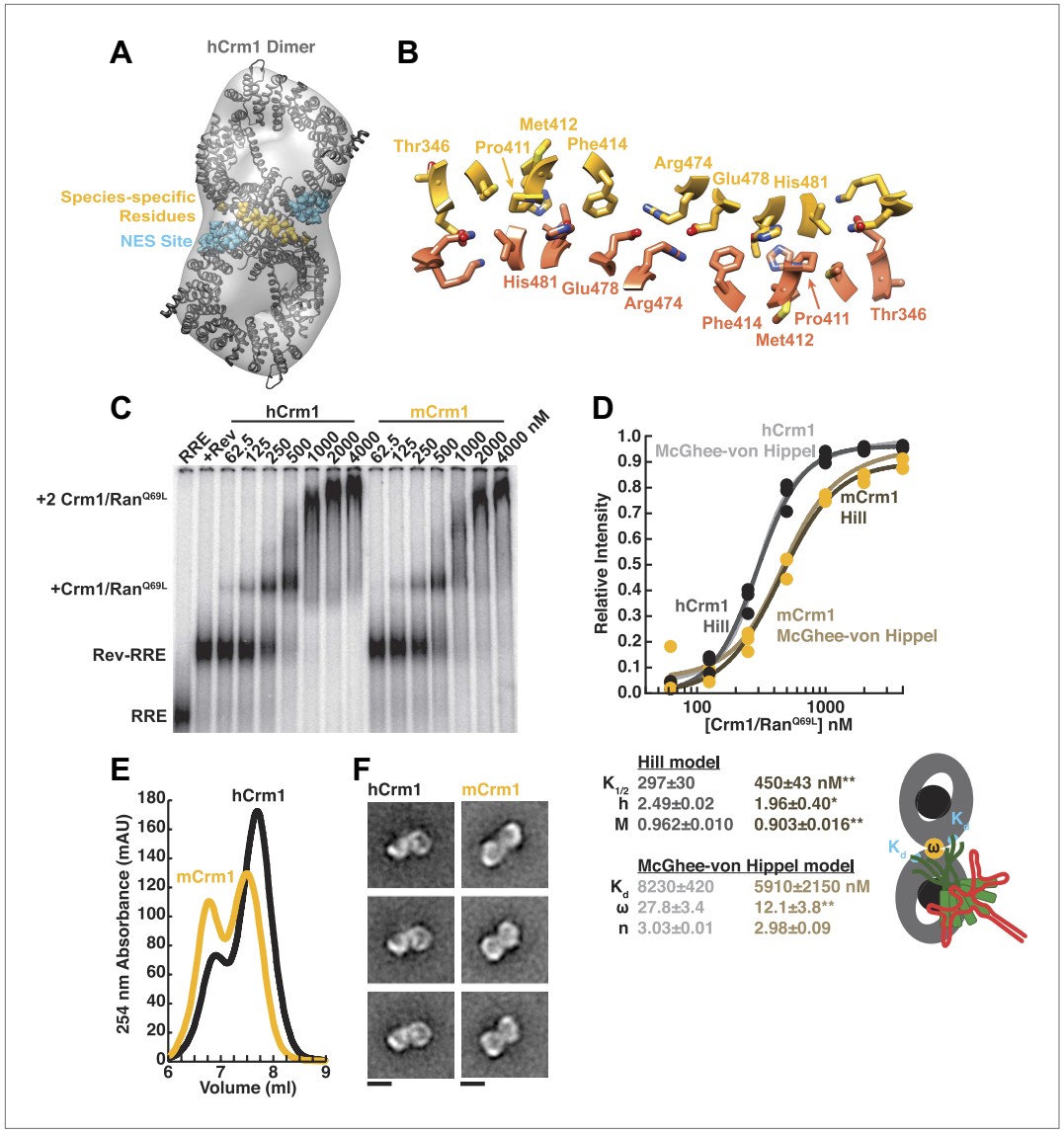

**Figure 2**. Species-specific residues define a Crm1 dimerization surface that enhances Rev-RRE binding. (**A**) A dimer of human Crm1 extracted from a unit cell of a Crm1-Snurportin1 complex that lacks Ran[GTP] (PDB 3GB8) was fit into the EM reconstruction (grey envelope) of Crm1 and Ran[GTP] bound to Rev-RRE (*Figure 2—figure supplement 2*). Residues that differ between murine and human Crm1 that enhance Rev-RRE activity (*Sherer et al., 2011*; *Elinav et al., 2012*) are shown in gold and form an interface between two Crm1 monomers (detailed in panel **B** and *Figure 2—figure supplement 3*). Two NES binding sites poised to engage two Rev NES peptides are shown in cyan. (**C–D**) Human Crm1 recognizes Rev-RRE with higher affinity and increased cooperativity compared to its murine ortholog, shown by gel mobility shift assays with radiolabeled RRE, a molar excess of Rev, and increasing concentrations of Crm1-Ran[GTP]. The Hill equation or a version of the McGhee-von Hippel model (*Epstein, 1978*) was fit to the quantified data from three independent experiments ($R^2 > 0.99$ for all models compared to average values) as shown in (**D**) and described in 'Materials and methods'. Asterisks denote *$p < 0.05$ and **$p < 0.01$ from one-tailed t-tests. A schematic model of the complex is shown to highlight the physical meaning of the McGhee-von Hippel parameters. (**E**) Size-exclusion chromatograms of Rev-RRE complexes assembled with murine Crm1 or human Crm1 show that murine Crm1 elutes with a larger apparent mass, presumably due to a larger hydrodynamic radius in the absence of Crm1-Crm1 interactions. (**F**) Class averages of Rev-RRE complexes bound to a dimer of human (426 particles) or murine (617 particles) Crm1 from negatively-stained micrographs of particles from (**E**). Murine Crm1 particles showed a wider range of sizes than the human particles, with most being larger, and the murine particles showed different features, such as an hourglass shape with fewer details in the center of the particle than human Crm1 particles. Scale bars show 10 nm.

*Figure 2. Continued on next page*

*Figure 2. Continued*

The following figure supplements are available for figure 2:

**Figure supplement 1**. Random conical tilt reconstruction and refinement.

**Figure supplement 2**. Comparison of fitting murine Crm1 or human Crm1 crystal structures in the EM reconstruction.

**Figure supplement 3**. A comparison of species-specific residues at the interface of the Crm1 dimer.

structure (*Nagai-Fukataki et al., 2011*; *Sherer et al., 2011*; *Elinav et al., 2012*). Based on steric considerations from the structure of a Rev oligomer, our lab previously proposed a 'jellyfish' model in which a Rev hexamer with dangling unstructured NES peptides could accommodate no more than two Crm1 subunits (*Daugherty et al., 2010b*). As described below, our data now suggest that Crm1 forms an intimate dimer with Rev-RRE bridging across the interface, permitting us to propose a more comprehensive model of the export complex.

A detailed examination of the Crm1 dimer reveals an extensive interface that buries a total surface area of 2170 Å² in the crystal structure (*Figure 2B*) and corresponds to functionally important residues (*Figure 2—figure supplement 3*). Central to the interface are symmetric hydrogen bonds between Arg474 and Glu478 across the dimer (2.9 Å donor to acceptor; *Figure 2B*) and, indeed, substituting these residues decreases Rev function (*Elinav et al., 2012*). The surrounding interactions are largely hydrophobic: Phe414 fills a cavity between Glu478 and His481 from the opposite subunit while Pro411, Met412, and Met 348 interact with the apposing Val484. Consistently, swapping murine residues for Pro411, Met412, and Phe414 results in the largest reduction in activity (*Sherer et al., 2011*; *Elinav et al., 2012*). The ends of the interface near Thr346 utilize Glu345 to form a salt bridge with Lys531 (2.8 Å) and hydrogen bond with Gln530 (2.8 Å) on the other monomer. The sum of these interactions enables human Crm1 to form an ordered dimer interface upon Rev-RRE binding.

## The Crm1 dimer interface facilitates cooperative assembly with the Rev-RRE complex

To assess the importance of this interface for Rev-RRE recognition, we compared the binding of murine and human Crm1 to Rev-RRE complexes using gel mobility shift assays (*Figure 2C*), which were quantified and fit with a standard Hill binding model (*Figure 2D*). The apparent binding constant for human Crm1 to Rev-RRE was slightly, yet significantly ($p < 0.01$), higher than for murine Crm1 ($K_{1/2} = 297 \pm 30$ nM vs $450 \pm 43$ nM), and the human protein achieved higher maximal saturation ($M = 0.96 \pm 0.010$ vs $0.90 \pm 0.016$, $p < 0.01$). An interesting but puzzling difference also was observed in the cooperativity term, as measured by the Hill coefficient ($h = 2.49 \pm 0.02$ vs $1.96 \pm 0.40$, $p < 0.05$). For human Crm1, the Hill coefficient exceeded the dimeric stoichiometry of the complex, which—while unusual—is not unprecedented (*Atkinson, 1966*). As an initial interpretation, we considered that the Hill coefficient of two with murine Crm1 reflected the stoichiometry of the complex while the higher value with human Crm1 might also include an additional component of cooperativity resulting from the arrangement of the Crm1 dimer. Overall, the result of these small differences between the human and murine proteins is comparable to the fourfold enhancement in nuclear export activity seen when human Crm1 is expressed in murine cells (*Sherer et al., 2011*; *Elinav et al., 2012*).

Because fitting to the Hill model yielded somewhat atypical parameters, we explored whether an alternative model, which describes how large ligands bind to a linear array of sites (*McGhee and von Hippel, 1974*), might better explain the data. This model has typically been used to analyze protein binding to DNA sites, which may resemble how Crm1 binds to the array of NES peptides in the Rev-RRE complex. According to this model, the affinity between the ligand and each binding site ($K_d$) would describe Crm1 binding to one Rev NES, the number of binding sites that a large ligand occludes (n) would reflect the number of Rev sites Crm1 obscures due to its large size, and cooperativity between ligands (ω) would be reflected by ω > 1, indicating an interaction between Crm1 monomers (*Figure 2D*). In this model, human and murine Crm1 both have similar affinities for an individual NES ($K_d \approx 6$–8 μM) but human Crm1 shows more than 2-fold higher cooperativity (ω = $27.8 \pm 3.4$ vs $12.1 \pm 3.8$, $p < 0.01$).

It is common for ω values to be quite large if complexes have extensive protein–protein interfaces, such as the binding of RecA filaments to DNA, with ω ≈ 50 (*Menetski and Kowalczykowski, 1985*). The relative ω values for Crm1 suggest a tighter interaction between the human Crm1 monomers, but the absolute value of ω > 1 for murine Crm1 still suggests some interaction between monomers. Indeed, murine Crm1 complexes with Rev-RRE elute from size-exclusion columns with an even larger apparent size than with human Crm1 (*Figure 2E*) and display a different, elongated, hour-glass shape in class averages of negatively-stained particles (*Figure 2F*), suggesting that these differences in the dimer complexes may reflect observed differences in Rev-RRE binding and export activities. Both human and murine Crm1 occlude three binding sites (*Figure 2D*), consistent with the stoichiometry of a Rev hexamer harboring six NESs binding to only a dimer of Crm1 (*Daugherty et al., 2010b*). These binding analyses also suggest that the Crm1 dimer interface helps promote cooperative assembly of the complex. The stabilization of the Rev-RRE export complex imparted by the human Crm1 dimer interface may explain how human Crm1 enhances oligomerization of a Rev-like protein from Human T-cell Leukemia Virus-1 (*Hakata et al., 2003*) or why it enhances the nuclear export activity of Rev for other lentiviruses, such as feline immunodeficiency virus (*Elinav et al., 2012*).

The micromolar affinities for Crm1 binding to an individual NES calculated with the McGhee-von Hippel model are consistent with previous reports (*Paraskeva et al., 1999*; *Güttler et al., 2010*) and underscore the importance of the RRE for scaffolding the Rev oligomer to promote the interaction with Crm1. For example, murine Crm1, with apparently weak or no interaction between Crm1 monomers, still binds the Rev-RRE complex with an order of magnitude greater affinity than to a single NES ($K_{1/2}$ = 450 nM vs $K_d$ = 5910 nM). This result may indicate that the increased local concentration of NESs in the Rev-RRE complex enhances Crm1 binding and is consistent with our observations in *Figure 1B* where the RRE was required for Rev to strongly bind Crm1. Furthermore, a recent study has shown that increasing the local concentration of NESs in the Rev-RRE complex by appending additional NESs to Rev overcomes the defect in murine Crm1 (*Aligeti et al., 2014*). Overall, the combined contributions from the RRE scaffolding the Rev oligomer and the interaction between Crm1 monomers cooperatively assemble the whole complex for nuclear export and may strongly influence the disassembly of the particle in the cytoplasm to regulate downstream steps for translating viral proteins and virion assembly (*Yedavalli et al., 2004*; *Nagai-Fukataki et al., 2011*; *Elinav et al., 2012*).

## Rev-RRE binds between NES binding sites on one side of the Crm1 dimer

To reveal how the Rev-RRE particle docks onto the export complex, we fit a dimer of Crm1 with bound Ran^GTP (*Figure 2—figure supplement 2*) into the EM map and then searched for extra density. Such extra density, which sits on one side of the Crm1 dimer between the two NES binding sites, is expected to correspond to bound Rev-RRE (*Figure 3A,B*). The location of this density sharply contrasts with how a host cargo, Snurportin1, binds to monomeric Crm1, where the main docking surface is on the opposite side of Crm1 (*Figure 3C*). Yet, the observed density is too small to fully accommodate a Rev hexamer bound to RNA, likely reflecting the poor visualization of RNA in uranyl-stained images as well as conformational heterogeneity (*Figure 3—figure supplement 1*) arising from the disordered carboxy-terminus of Rev where the NES is located (Daugherty et al., 2010).

To be certain of the locations of Rev and the RRE in the particle, we compared class averages of the complex to those formed either with GB1-tagged Rev or with streptavidin-tagged RRE (*Figure 3D*). Despite heterogeneity arising from flexible linkers to the tags and from different conformations within each class, additional density was plainly visible in both tagged complexes (*Figure 3—figure supplement 2*) with Rev and the RRE unambiguously positioned on one side of the Crm1 dimer. Class averages from cryo-electron micrographs, where RNA staining is not an issue, also show extra density in the same location (*Figure 3D*). Together these data indicate that the positioning of NES sites constrains Rev-RRE to flexibly dock on one side of the Crm1 dimer.

## Discussion

This work provides the first glimpse of an assembled HIV nuclear export complex, involving an essential HIV-host interaction between the Rev-RRE complex and Crm1. Most surprisingly, we revealed the existence of a Crm1 dimer, whose details could be inferred from crystal contacts observed in an existing Crm1 structure (*Dong et al., 2009*). The ability to assemble these complexes in vitro provides a good starting point to investigate how additional host proteins may work with Crm1 during viral RNA trafficking.

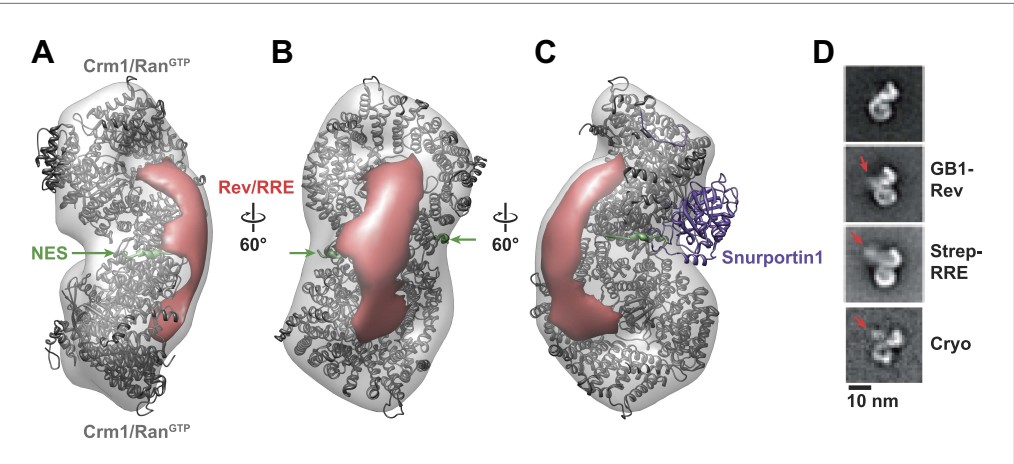

**Figure 3**. Rev-RRE binds between NES binding sites on the Crm1 dimer. (**A–B**) A model of the Crm1-Ran^GTP dimer displays extra density (red) on one side of the structure that locates the Rev-RRE complex between the Rev NESs bound to each Crm1 (green). The Crm1-Ran^GTP dimer model was built by aligning two murine Crm1-Ran^GTP complexes onto the human Crm1 dimer. (**C**) A rotated view of the structure showing the location of Snurportin1 bound to a Crm1 monomer on the opposite side of the protein. (**D**) Class averages of particles containing Rev tagged with GB1 or RRE tagged with streptavidin, and cryo-EM of the complex without any tags, confirm the location of Rev-RRE. Raw micrographs, class averages, and particle numbers are shown in *Figure 3—figure supplement 2*.

The following figure supplements are available for figure 3:

**Figure supplement 1**. Heterogeneity within class averages reveals Rev-RRE density.

**Figure supplement 2**. Localization of Rev-RRE density.

The Rev-RRE particle is clearly positioned at the Crm1 dimer interface although the extent of the interface between Rev-RRE and Crm1 is unclear beyond the Rev NES interaction because there was insufficient resolution to precisely place Rev-RRE into the reconstructed complex. In addition to poor staining of the RNA, heterogeneity of the Crm1/Rev-RRE interaction contributes to the lack of distinct density for Rev-RRE and presumably results from multiple arrangements of the complex as well as peptide flexibility surrounding the Rev NESs. Furthermore, models of Rev oligomers using combinations of Rev dimer structures free or bound to RNA suggest that a range of oligomer conformations can interface with the Crm1 dimer (*Jayaraman et al., 2014*). Determining which conformations are present in the entire export complex will require higher resolution structures.

Because a portion of the RRE stabilizes an alternate conformation of the Rev dimer (*Jayaraman et al., 2014*) and the whole RRE directs Rev oligomerization (*Pond et al., 2009*; *Fang et al., 2013*; *Bai et al., 2014*), we anticipate that the RNA modulates the conformation of the Rev oligomer and consequently its interaction with the Crm1 dimer. Indeed, we observe that the RRE enhances the interaction with Crm1, probably reflecting the high local concentration of Rev NESs promoted by the RRE scaffold. If the RRE stabilizes distinct conformations of the Rev oligomer (*Jayaraman et al., 2014*), then the volume occupied by the Rev NES in those different states could influence the affinity for the Crm1 dimer. Such a mechanism could explain how HIV can evolve to fine-tune the levels of nuclear export through changes in the RRE during the course of infection within a single host (*Legiewicz et al., 2008*; *Sloan et al., 2013*).

It is clear that the Crm1 dimer architecture is functionally important for HIV replication since mutations in Crm1 that decrease Rev-dependent export and restrict viral replication correspond to residues that form the interface between Crm1 monomers (*Nagai-Fukataki et al., 2011*; *Sherer et al., 2011*; *Elinav et al., 2012*). The difference between murine and human Crm1 amplifies into a fourfold defect in nuclear export and a similar defect in translation of viral proteins (*Figure 4*). Overall, these subtle differences in binding lead to a 12-fold increase in virion production when human Crm1 is expressed in mouse cells (*Sherer et al., 2011*), and even these values may underestimate the differences in an

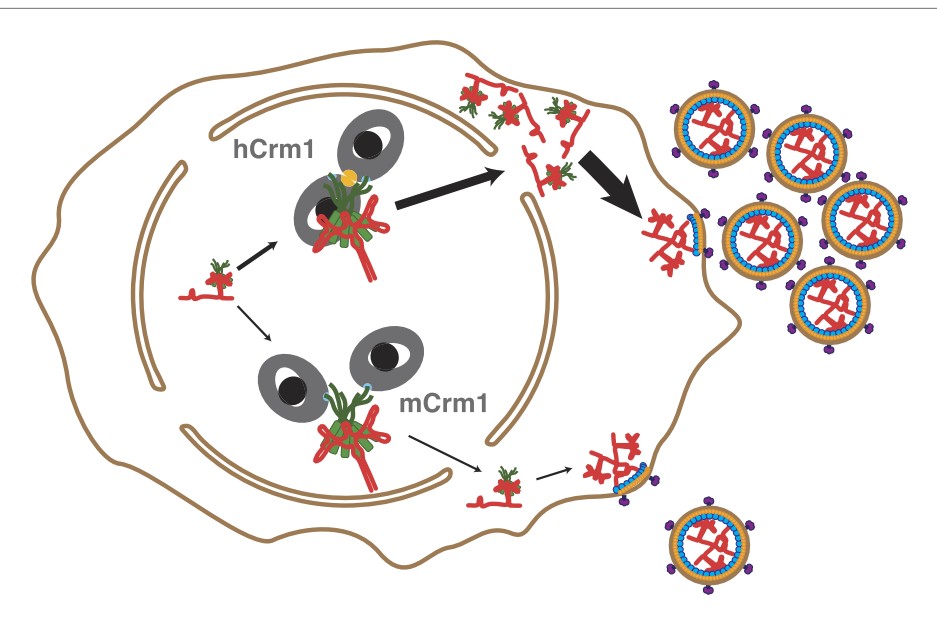

**Figure 4**. Cooperative assembly of a nuclear export complex with human Crm1 enhances nuclear export and virion production. Human Crm1 (top) associates with Rev-RRE with higher affinity than murine Crm1 (bottom) due to species-specific residues that form the dimer interface (gold). This increased association amplifies into even larger differences in nuclear export activity and virion assembly between the two Crm1s.

endogenous context since in these experiments human Crm1 is overexpressed in cells already containing murine Crm1.

The significance of the Crm1 dimer for Rev-RRE function is underscored by a comparison of Crm1 orthologs (*Sherer et al., 2011*). This work proposed an intriguing evolutionary path for Crm1 starting from an ancestral sequence that lacked the full set of residues that we now know form the Crm1 dimer interface. The simian primate lineage might have acquired mutations that stabilize the interface, whereas the murine lineage might have accrued mutations that destabilize the interface. Even though mouse Crm1 lacks the residues to fully stabilize the Crm1 dimer, the Rev-RRE complex can still recruit two Crm1 subunits, which presumably accounts for its partial function in Rev-mediated export. Thus, Rev oligomerization scaffolded by the RRE may have initially allowed ancestral viruses to recruit two Crm1 subunits even if an ancestral form of Crm1 did not have a dimeric interface. The oligomerization of Rev-like proteins in other retroviruses may similarly facilitate binding to existing Crm1 orthologs that lack the dimeric interface. The selective pressures that fixed the dimer interface in the simian primate lineage are unknown, but its evolution has certainly provided an advantage for HIV replication.

The Crm1 surface bound by Rev-RRE is clearly distinct from where the host cargo Snurportin1 binds (*Figure 3C*) and could reflect a virus-specific interface and potential therapeutic target. However, it seems more likely that the Crm1 dimer plays some role in host cell biology. Previous studies have implicated the same species-specific Crm1 residues as important for binding to RanBP3, a protein that enhances interactions of Crm1 with Ran$^{GTP}$ and some NESs, although it is not known if a Crm1 dimer is involved (*Hakata et al., 2003*). Alternatively, Crm1 dimerization may modulate levels of nuclear export for host mRNAs. Crm1 mediates export of a small set of mRNAs via proteins bound near their 3′ ends (*Brennan et al., 2000*; *Yang et al., 2001*; *Culjkovic et al., 2006*; *Prechtel et al., 2006*). One of these proteins, HuR, even forms an oligomeric complex at its 3′ RNA element (*Fialcowitz-White et al., 2007*). Perhaps analogous to the Rev, the HuR oligomer may also bind a Crm1 dimer. Moreover, the positioning of cis-acting elements within a cellular mRNA may define a topology for RNA-binding proteins to recruit a Crm1 dimer without homooligomeric assemblies of proteins. Such a mechanism would allow Crm1-mediated export to integrate different protein signals to tune the levels of nuclear export for a given mRNA.

## Materials and methods

### Molecular cloning

All plasmids used in this study are listed in *Supplementary file 1*. Crm1 and Ran were cloned between the NdeI and XhoI sites in a pET19b vector (EMD Millipore, Darmstadt, Germany) that encodes an amino-terminal deca-histidine tag followed by a tobacco etch virus (TEV) protease site. Genes were amplified from their source vectors by polymerase chain reaction with phusion polymerase (New England Biolabs (NEB), Ipswich, MA) using forward and reverse primers (Integrated DNA technologies (IDT), Coralville, IA, listed in *Supplementary file 2*) containing NdeI or XhoI restriction sites, respectively. PCR products and the target vector were digested with restriction enzymes (NEB), purified (Qiagen, Venlo, Netherlands), ligated (Roche, Basel, Switzerland), and then transformed into *E. coli DH5α*. Plasmids from positive transformants were isolated (Qiagen), screened by restriction digests, and sequenced (University of California - Berkeley DNA Sequencing Facility, Berkeley, CA).

### Site-directed mutagenesis

Mutations were introduced using a single primer method (*Makarova et al., 2000*). Briefly, DNA was synthesized using mutation-bearing primers (IDT) using Pfu Turbo (Agilent, Santa Clara, CA), purified, DpnI digested, and transformed into *E. coli DH5α*. Plasmids from transformants were isolated and sequenced.

### RNA Transcription and purification

RNA was transcribed by T7 polymerase from a linearized plasmid and then purified by Urea-PAGE (*Daugherty et al., 2008*). A single-stranded segment found to be a cloning byproduct at the 5′-end of the RRE was removed from the original template by site-directed mutagenesis. Pure RNA pellets were resuspended in buffer A [1 mM HEPES-KOH, pH 7.5 @ 20°C] and stored at −20°C. RNA was annealed by heating to 95°C and cooling to 30°C in an aluminum heating block placed at room temperature. RNA was quantified by UV spectroscopy using a 260 nm extinction coefficient of $2.5 \times 10^6$ M$^{-1}$cm$^{-1}$ for the RRE.

### 5′-end labeling RNA

Purified RNA was treated with calf intestinal phosphatase according to the manufacturer (NEB). RNA was purified by phenol-chloroform extraction, ethanol precipitated in 300 mM NaCH$_3$CO$_2$, washed with 70% ice-cold ethanol, and resuspended in buffer A. RNA was labeled for ≥1 hr at 37°C in buffer B [50 mM imidazole-HCl, pH 6.4 @ 20°C, 10 mM MgCl$_2$, 5 mM DTT] with 10 Units of Polynucleotide Kinase (NEB) by adding either γ-$^{32}$P-ATP (Perkin Elmer, 3000 Ci/mmol) or ATPγS (Sigma-Aldrich, St. Louis, MO) to final concentrations of 450 nM and 1 mM, respectively. For radiolabeling, RNA was purified by phenol chloroform extraction, diluted in buffer A, and desalted with PD Spintrap G-25 column (GE Healthcare Lifesciences, Piscataway, NJ) according to manufacturer's instructions. For biotin labeling, RNA was purified by adding NH$_4$CH$_3$CO$_2$ to a final concentration of 2.5 M and centrifuging at >20,000×$g$ and 4°C for 1 hr to pellet the protein. RNA in the supernatant was ethanol precipitated and resuspended in buffer A. Maleimido-biotin was added at a final concentration of 100 μM, incubated at room temperature for 4 hr, and quenched with an excess of 2-mercaptoethanol. The reaction was diluted in buffer A, desalted as above, and ethanol precipitated twice in NaCH$_3$CO$_2$ in order to get rid of excess label. The washed RNA pellet was resuspended in buffer A. After labeling, RNAs were annealed as above and evaluated on Urea-PAGE to ensure no degradation. Radiolabeled RNA was quantified by scintillation counting, aliquoted, and then stored at −20°C.

### Protein expression

Proteins were expressed in *E. coli* BL21(DE3). Bacteria were grown in Luria–Bertani Miller Formula (LB) media with noted supplements by diluting overnight cultures 50-fold into 1 l of media and by shaking at 200 rpm in baffled Fernbach flasks to the indicated OD$_{600}$. Expression was induced by adding isopropyl-β-D-thiogalactopyranoside (IPTG) to the indicated final concentration. Bacteria were harvested by centrifugation at 4500×$g$ and 4°C for 20 min. Pellets were frozen in N$_2$ (*l*) and stored at −80°C.

### Rev expression

Rev was expressed as previously described (*Daugherty et al., 2008*).

## Crm1 expression

LB media and agar plates were supplemented after autoclaving with 1% (wt/vol) glucose, 100 µg/ml Ampicillin, and 34 µg/ml chlorampenicol. Crm1 expression vectors were co-transformed with pLysS-Rosetta2 (EMD Millipore), and colonies were selected on LB agar plates. Bacteria were grown for protein expression with chloramphenicol reduced to 17 µg/ml at 37°C until $OD_{600}$ = 1.0–1.2. Cultures were shifted to 20°C for 20 min, induced by adding IPTG to 200 µM, and grown overnight.

## Ran expression

Bacteria were grown in LB with 100 µg/ml Ampicillin at 37°C to $OD_{600}$ = 0.6-0.8, shifted to 18°C for 20 min, induced with 500 µM IPTG, and grown overnight.

## Rev purification

Each gram of bacterial pellet was resuspended in 10 ml of buffer C [50 mM Tris–HCl, pH 8.0 @ 20°C; 2 M NaCl; 2 mM 2-mercaptoethanol; 10 mM imidazole; 0.1% (vol/vol) Tween-20]. Proteases were inhibited by adding phenylmethanesulfonylfluoride (PMSF) to a final concentration of 1 mM and one complete, Mini, EDTA-free protease inhibitor cocktail tablet (Roche) was added for every 5 g of bacteria. Bacteria were lysed with an Emulsiflex-C3 homogenizer (Avestin, Ottawa, Canada), and the lysate was cleared by centrifugation at 30,000×$g$ and 4°C for 30 min. The supernatant was loaded on Ni-NTA Agarose resin (Qiagen) with 1 ml of resin for every 2.5 g of bacterial pellet. The resin was washed extensively with buffer C and then buffer D [125 mM $K_2HPO_4$-$KH_2PO_4$, pH 7.5 @ 20°C; 250 mM NaCl; 2 mM 2-mercaptoethanol; 10% (vol/vol) glycerol; 0–500 mM imidazole]. A step-wise increase in imidazole concentration eluted proteins from the resin. Pure fractions were combined, supplemented with TEV protease at a 1:20 molar ratio, and dialyzed overnight in a 3500 Da MWCO Slide-A-Lyzer Cassette (Thermo Scientific, Waltham, MA) against buffer D without imidazole. Dialyzed protein was passed over Ni-NTA resin. The flow-through was collected and filtered through a 0.22 µm syringe filter. The protein was then loaded onto a Mono-S 5/50 GL column (GE Healthcare Lifesciences), washed, and eluted with a linear gradient of buffer D into buffer E [500 mM $K_2HPO_4$-$KH_2PO_4$, pH 7.5 @ 20°C; 1 M NaCl, 2 mM 2-mercaptoethanol; 10% (vol/vol) glycerol]. After analysis by SDS-PAGE, pure fractions were pooled, frozen in $N_2$ (l) and stored at −80°C. Rev concentration was determined by UV spectroscopy using 280 nm extinction coefficients of 20,000 $M^{-1}cm^{-1}$ for GB1-Rev and 8480 $M^{-1}cm^{-1}$ for Rev.

## Crm1 purification

Crm1 was resuspended in 4 ml of buffer F [50 mM Tris–HCl, pH 7.5 @ 4°C; 600 mM NaCl; 2 mM $Mg(CH_3CO_2)_2$; 2 mM 2-mercaptoethanol; 10-200 mM imidazole; 20% (vol/vol) glycerol] containing 0.1% (v/v) Tween-20 and 5% glycerol for each gram of bacterial pellet. A protease inhibitor tablet was added for every 40 g of bacteria with PMSF. Lysis was as for Rev. Contaminating nucleic acids were removed from the supernatant of the clarified lysate by adding polyethylenimine to 0.2% (wt/vol) and cleared by centrifugation at 20–25,000×$g$ and 4°C for 30 min. Ni-NTA chromatography was as for Rev with with 1 ml of resin for every 20 g of bacterial pellet and washes and elutions were in buffer F. TEV protease was added to pooled fractions in a 1:20 molar ratio and incubated at 4°C overnight. Protein was concentrated with a 10 kD MWCO Amicon Ultra Free Centrifugal Concentrator (EMD Millipore) and filtered through a 0.22 µm syringe filter. Crm1 was further purified by size-exclusion chromatography with buffer F containing 10% glycerol and 20 mM imidazole on a Superdex 200, HiLoad 16/60 column (GE Healthcare Lifesciences) followed in tandem by a HisTrap Fast Flow, 5 ml column (GE Healthcare Lifesciences). Fractions were analyzed by SDS-PAGE. Pure fractions were pooled, concentrated, frozen in $N_2$ (l), and stored at −80°C. Prior to use Crm1 was exchanged into buffer G [50 mM HEPES-KOH, pH 7.5 @ 20°C, 50 mM KCl, 2 mM $Mg(CH_3CO_2)_2$, 2 mM 2-mercaptoethanol, 2% (vol/vol) glycerol] using a 7 kD MWCO, Zeba Spin column (Thermo Scientific), and the concentration was determined using Bradford assays with a bovine serum albumin (BSA) standard (Thermo Scientific).

## Ran purification and nucleotide exchange

Ran was purified in a similar manner as Rev with the following modifications: Lysis, Ni-NTA purification, and dialysis were performed with buffer H [64 mM Tris–HCl, pH 7.5 @ 4°C, 400 mM NaCl, 5 mM $MgCl_2$, 2 mM 2-mercaptoethanol, 0–250 mM Imidazole]. Prior to cation exchange, protein was concentrated with a 3 kD MWCO Amicon Ultra Free Centrifugal Concentrator (EMD Millipore), nucleotides were removed by adding EDTA to a final concentration of 10 mM, and exchanged into

buffer I [25 mM MES-HCl, pH 6.5 @ 4°C, 0.020–1 M KCl, 1 mM EDTA, 4 mM 2-mercaptoethanol, 10% (vol/vol) glycerol] using a PD-10 column (GE Healthcare Lifesciences). Any precipitate was removed by centrifugation at 20,000×$g$ followed by filtration through a 0.22 μm syringe filter. Cation exchange was performed with buffer I and eluted with a linear gradient of KCl in buffer I. Prior to use, Ran nucleotides were exchanged into Ran by adding nucleotide and MgCl$_2$ to final concentrations of 1 mM and 5 mM, respectively, and then incubating at room temperature for 30 min. Excess nucleotide was removed by exchanging the protein into buffer G using a 7 kD MWCO, Zeba Spin Column (Thermo Scientific). Protein concentration was determined with a BSA standard as for Crm1.

### In vitro pulldowns

Ran was exchanged with GTPγS or GDP as described above. Rev was diluted with buffer G containing 30 mM imidazole, and Crm1 and Ran were exchanged into the same buffer in order to reduce non-specific interactions with the resin.1.6 μM GB1-Rev was mixed with 200 nM RRE, 400 nM Crm1, and/or 800 nM Ran in 50 μL. Reactions were incubated with 50 μl of Ni-NTA agarose resin for 2 hr at 25°C. The supernatant was removed and the beads were washed three times with 500 μl of buffer. Proteins were eluted by rotating the resin for 5 hr with 100 μl of assembly buffer containing 750 mM imidazole. Eluates were separated on a 4–12% Novex Bis-Tris Gel in SDS-MES buffer (Life Technologies, Waltham, MA) and stained using SYPRO Ruby (Lonza, Basel, Switzerland). In these gels, only Rev and Crm1 are easily visible because Rev and Ran migrate with the same apparent mass.

### Size-exclusion chromatography and multi-angle laser light scattering (MALLS)

Rev-RRE complexes were assembled with a 10-fold excess of Rev added to the RRE. Crm1 and Ran$^{Q69L}$ were then added to Rev-RRE and diluted to 70 μl. The final concentration of the Rev-RRE complex was 4 μM for stoichiometric titrations. At these concentrations, Rev-RRE completely assembles with Crm1 and Ran$^{Q69L}$ but disassembles through dilution on the column, as supported by native gel electrophoresis (*Figure 2B*). The assembly reactions were incubated at 37°C, filtered through a 0.22 μm centrifugal filter (EMD Millipore), and then kept on ice. A Shodex KW-804 (Showa Denko America, New York, NY) column equilibrated in buffer G was loaded with 50 μl of assembly reaction. Complexes were separated at a flow rate of 0.35 ml/min, and the 12.5 ml column volume was collected in 135 μl fractions. Fractions were analyzed by SDS-PAGE as for in vitro pulldowns. Molecular masses were determined using the program ASTRA from MALLS and refractometry (Wyatt Technology Corp, Santa Barbara, CA) measurements following size-exclusion chromatography. For these experiments, the concentration of Rev-RRE was increased to 8 μM for accurate mass determination. The masses of Crm1 and Ran$^{Q69L}$ were determined at 20 μM and 50 μM, respectively. Because the technical error of measuring the mass of the nuclear export particle was nearly the same as a Rev monomer, we validated the stoichiometry by densitometry of SDS-PAGE gels stained with Sypro-Ruby scanned on a Typhoon (GE Healthcare Lifesciences) and quantified with ImageQuant. Concentrations for standards of each protein were determined by amino-acid analysis (UC-Davis Proteomics Facility, Davis, CA).

### Gel mobility shift assays

Crm1 and Ran were exchanged into buffer J [50 mM HEPES-KOH, pH 7.5 @ 20°C, 100 mM KCl, 2 mM Mg(CH$_3$CO$_2$)$_2$, 0.5 mM EDTA, 2 mM 2-mercaptoethanol, 12% (vol/vol) glycerol] and protein concentrations were determined by quantitative SDS-PAGE. Stocks of Rev and the RRE were diluted in buffer J and then combined at 80 nM and 2000 cpm/μl (~0.5 nM), respectively. Crm1 and Ran$^{GTP}$ were serially diluted in buffer J containing 100 μg/ml BSA, 10 μM yeast tRNA, and 200 nM methionine. 6 μl of Crm1-Ran$^{GTP}$ was added to 6 μl of Rev-RRE and assembled at 37°C for 15 min. 10 μl of the assembly reaction was loaded onto a 5% acrylamide (Protogel, 37.5 acrylamide to 1 bisacrylamide) native gel in buffer K [25 mM HEPES-25 mM imidazole, pH 7.4 @ 20°C] and continuously running at ~10 V/cm and 20°C. The gel was dried onto filter paper, exposed to a phosphor-screen overnight, and imaged on a Typhoon.

Band intensities (I) were quantified using ImageQuant and normalized with the equation,

$$I_{norm} = 1 - \left(\frac{I_j^{Rev-RRE}}{I_j^{Total}}\right) \bullet \left(\frac{I_o^{Total}}{I_o^{Rev-RRE}}\right)$$

where $I_j$ is the intensity at the indicated concentration of Crm1-Ran$^{GTP}$ and $I_o$ is the initial intensity of Rev-RRE without Crm1-Ran$^{GTP}$. A modified Hill equation,

$$I_{norm} = M \cdot \frac{C^h}{C^h + K_{1/2}^h}$$

where $K_{1/2}$ is the apparent binding constant, h is the Hill coefficient, M is the maximum instensity, and C is the concentration of Crm1 and Ran$^{Q69L}$, was fit to each binding experiment using sum-of-squares minimization implemented with the Solver tool in Microsoft Excel.

A version of the McGhee-von Hippel binding model (*McGhee and von Hippel, 1974*) derived for a finite lattice (*Epstein, 1978*) was also used to fit the data with a sum-of-squares minimization (*Kowalczykowski et al., 1986*). The model is described by the equation,

$$\theta = \frac{n}{L} \cdot \left( \frac{\sum_{k=0}^{L/n} \sum_{j=0}^{k-1} k \cdot P_M(k,j) \cdot \left(K_d^{-1}C\right)^k \omega^j}{\sum_{k=0}^{L/n} \sum_{j=0}^{k-1} P_M(k,j) \cdot \left(K_d^{-1}C\right)^k \omega^j} \right)$$

Where,

$$P_L(k,j) = \frac{(L-nk+1)!\,(k-1)!}{(L-nk-k+j+1)!\,(k-j)!\,j!\,(k-j-1)!}$$

$\theta$ is the fraction bound and is assumed to be the same as $I_{norm}$, L is the maximal number of binding sites, $K_d$ is the dissociation constant for Crm1 binding to a Rev NES, n is the number of sites occluded by Crm1 binding, $\omega$ is the cooperative interaction between Crm1 monomers, C is the concentration of Crm1/Ran, k is the number of Crm1 monomers bound to Rev-RRE, j is the number of times Crm1 monomers are adjacent to each other, and Pl(k,j) is number of combinations for k monomers binding to Rev-RRE with j adjacencies on a lattice of length L. In order to obtain reliable fits, L was set to an arbitrarily large number of 100 binding sites, which effectively describes an infinitely long lattice (*McGhee and von Hippel, 1974*; *Epstein, 1978*). Such a lattice can serve as a valid model for even a small number of binding sites (*Epstein, 1978*; *Kowalczykowski et al., 1986*). Because L is constant and the parameters k and j are determined from L and n, the only free parameters that vary are K, n, and $\omega$. We initiated the parameter search by setting $K_d$ = 1 μM, $\omega$ = 1, and n = 3. We additionally tested n = 1, 2, 3, or 4. n = 3 provided the most consistent fits with the best correlation, as measured by $R^2$, between the model and each of the titration curves.

## Negative-stain electron microscopy

2.5 μl of 100–400 nM samples were adsorbed to glow-discharged, carbon-coated, 400 mesh grids and stained with 0.75% uranyl formate as previously described (*Booth et al., 2011*). For random conical tilt, 200 mesh grids were used. A substoichiometric amount of streptavidin was added to biotinylated RNA immediately before adsorbing onto grids. Images were acquired at 1.5 μm defocus for untilted images and ~2.0 μm for tilted images using low dose procedures on a T12 microscope (FEI, Eindhoven, Netherlands) operating at 120 kV and recorded on a 4096 x 4096 CCD camera (Ultrascan 4000, Gatan, Pleasanton, CA) with a 52k magnification corresponding to a pixel size of 2.02 Å. Tilt pairs were manually collected at 60° and then 0° angles.

## Cryo-electron microscopy

2 μl of sample was applied to 1.2/1.3 μm Quantifoil Grids (Electron Microscopy Sciences, Hatfield, PA), blotted, and frozen in liquid ethane with a Vitrobot (FEI). Images were recorded with a 2–4.5 μm defocus range using low dose procedures on a TF20 microscope (FEI) operating at 200 kV and recorded on a 8192 × 8192 CMOS camera (TVIPS, Gauting, Germany) at 62k magnification corresponding to a pixel size of 1.29 Å.

## Image processing

Negative-stain images were binned twofold for a pixel size of 4.04 Å. Cryo-images were binned fourfold for a final pixel size of 5.17 Å. The defocus astigmatism, tilt angle, and tilt axis were determined using CTFFind and CTFTilt (*Mindell and Grigorieff, 2003*). Tilted images were rotated to align the tilt

axis with untilted images. Particles were manually selected using Samviewer (*Liao et al., 2013*). The tilt angle for each pair was further refined based on particle positions. Particles were boxed from images, combined into a single stack, normalized, and contrast transfer function (CTF) corrected. For cryo images, the contrast was inverted to have white particles against a black background as required for SPIDER (*Shaikh et al., 2008*). Particles from untilted negative-stain and cryo-images were subjected to reference free classification followed by multiple rounds of correspondence analysis, k-means classification, and multi-reference alignment using SPIDER (*Shaikh et al., 2008*; *Liao et al., 2013*).

## 3D reconstruction

For each class average of untilted images, we calculated a random conical tilt (RCT) reconstruction from tilted particles with angular parameters from the in plane rotation of class averages and from the tilt angle determined during particle picking. This initial RCT reconstruction was first refined for translational alignment of individual particles. Images of untilted particles were then added to the reconstructions followed by translational alignment. All refinements were performed using FREALIGN (*Grigorieff, 2007*) in a frequency-limited range of 240–38 Å. Volumes were low-pass filtered after each cycle of refinement using the FCREF option. When we compared RCT reconstructions using UCSF Chimera, we concluded the first five classes were different views of the same structure and the remaining two classes were substoichiometric particles (*Figure 2—figure supplement 1*). Untilted and tilted particles from the first five classes were merged into a single dataset for further projection matching refinement. The RCT reconstruction from Class02 was used as an initial model to determine Euler angles and in-plane shifts of all particles. In the subsequent iterative refinement, the frequency was limited between 240 Å and 34 Å. The final resolution was estimated to be 25 Å (FSC = 0.143) to 30 Å (FSC = 0.5) (*Figure 2—figure supplement 1*). The Euler angle distribution of all particles suggests there is no major gap in Euler space. A b-factor of −1000 Å$^2$ with a cosine-edge, low-pass filter to 25 Å and a 2-pixel width was applied to the final volume using b-factor (N. Grigorieff). We also tested applying C2 symmetry during the refinement and reconstruction. Symmetry improves the resolution of the overall reconstruction but weakens the features from Rev-RRE. We, therefore, only interpreted the volume with no symmetry applied.

## Molecular modeling

All molecular modeling was performed using UCSF Chimera (*Goddard et al., 2007*). Crystal structures were initially fit by eye into the EM density and further optimized using the 'Fit in Map' function in Chimera to optimize the correlation between the crystal structure filtered to 25 Å resolution and the EM map. In order to find the difference density for Rev-RRE, the hybrid model of the Crm1 dimer crystal structure (*Figure 2—figure supplement 2*) filtered to 25 Å resolution was used to mask out Crm1 and Ran density in the EM reconstruction using the 'mask' command in Chimera.

## Acknowledgements

We thank B Adams, B Linhares, I Ribeiro, and N Yonis for help with protein expression, M Braunfeld, X Li, M Liao, and S Wu for help with EM data collection and image processing, and D Agard and lab for help with MALLS. We also thank M Malim and I Mattaj for plasmids. We are grateful to B Jayaraman, Y Bai, J Doudna, M Malim, J Abelson, M Daugherty, C Guthrie and lab, P Walter, J Gross, C Gross, D Agard and lab, L Beamer, J Fraser, and members of the Frankel and Cheng Labs for helpful discussions, sharing data, and comments on the manuscript. This work was supported by NIH grant P50GM088250 to ADF and YC, and NIH grant R25GM56847 and NIH Ruth L. Kirchstein Predoctoral Fellowship F31GM095350 to DSB.

## Additional information

### Funding

| Funder | Grant reference number | Author |
|---|---|---|
| National Institute of General Medical Sciences | P50GM088250 | Yifan Cheng, Alan D Frankel |
| National Institute of General Medical Sciences | F31GM095350 | David S Booth |

| Funder | Grant reference number | Author |
|---|---|---|
| National Institute of General Medical Sciences | R25GM56847 | David S Booth |

The funders had no role in study design, data collection and interpretation, or the decision to submit the work for publication.

## Author contributions

DSB, Conception and design, Acquisition of data, Analysis and interpretation of data, Drafting or revising the article; YC, ADF, Conception and design, Analysis and interpretation of data, Drafting or revising the article

# Additional files

## Supplementary files

• Supplementary file 1. Plasmids for protein and RNA expression.

• Supplementary file 2. Primers.

## Major datasets

The following previously published datasets were used:

| Author(s) | Year | Dataset title | Dataset ID and/or URL | Database, license, and accessibility information |
|---|---|---|---|---|
| Dong X, Biswas A, Suel KE, Jackson LK, Martinez R, Gu H, Chook YM | 2009 | Crystal structure of CRM1/Snurportin-1 complex | http://www.pdb.org/pdb/explore/explore.do?structureId=3GB8 | Publicly available at RCSB Protein Data Bank. |
| Guttler T, Madl T, Neumann P, Deichsel D, Corsini L, Monecke T, Ficner R, Sattler M, Gorlich D | 2010 | Crystal structure of the HIV-1 Rev NES-CRM1-RanGTP nuclear export complex (crystal II) | http://www.pdb.org/pdb/explore/explore.do?structureId=3NC0 | Publicly available at RCSB Protein Data Bank. |
| Booth DS, Cheng Y, Frankel AD | 2015 | The export receptor Crm1 forms a dimer to promote nuclear export of HIV RNA | EMD-6231; http://emsearch.rutgers.edu/atlas/6231_summary.html | Publicly available at EMDataBank. |

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
