## [Decision Letter]

Thank you for sending your work entitled “HIV utilizes a Crm1 dimer to promote RNA nuclear export” for consideration at *eLife*. Your article has been favorably evaluated by John Kuriyan (Senior editor), Wes Sundquist (Reviewing editor), and 2 reviewers, one of whom, Jamie Williamson, has agreed to share his identity.

The authors describe the reconstitution from pure recombinant components and an EM reconstruction of the HIV-1 RNA export complex: i.e., a complex formed between the HIV-1 RRE and Rev, Ran(GTP), and the human exportin, Crm1. The authors clearly demonstrate that all of these components are required for efficient complex formation and they use a mutant Rev to show that the necessary interaction between Crm1 and Rev NES is indeed taking place. They then go on to present the surprising, but compelling argument that there are two copies of hCrm1 in the complex, rather than the expected single copy, even though Crm1 is monomeric in solution. The presence of a Crm1 dimer is shown both biochemically and by EM. In the EM work, a nominal ∼30 Å resolution reconstruction is generated by negative-stain single-particle electron microscopy, using the random conical tilt method to achieve a suitable distribution of viewing angles. The authors can fit a dimer extracted from the lattice of the hCrm1 crystal structure into their reconstruction, even though the intermolecular lattice contact had previously been assumed to be a crystallization artifact. Nicely, the human Crm1 dimer interface also appears to explain, at least in part, why HIV-1 Rev uses human Crm1 better than murine Crm1 (where the dimer interface is missing). The RRE-Rev complex binds more tightly and with greater cooperativity to the human Crm1-RanGTP complex vs. the murine complex (although the differences are modest), and residues that appear to dictate the species specificity of HIV-1 Rev map to the hCrm1 crystallographic dimer interface. Finally, the authors propose a model for the overall architecture of the HIV-1 export complex and posit that the spacing between the Rev binding sites on the Crm1 dimer are optimally distanced to recognized the C-terminal tails of the two terminal Revs in a hexameric complex.

Despite the relatively low resolution of the EM reconstruction, this paper is important because it reveals a new functional oligomeric state for hCrm1 (which may also be used in the export of larger host RNA complexes) and because it provides a satisfying molecular explanation for the well-known inability of HIV-1 Rev to utilize murine Crm1 efficiently. Although Crm1 has been previously characterized as binding other export cargoes as a monomer, the authors argue that Rev-RRE stabilizes the formation of a Crm1 dimer and that the dimer requirement reflects the greater size of this export cargo. The basic observations presented in this paper are an important advance in thinking about the Rev-RRE-Crm1 interactions, and how the HIV (and other) export complexes are formed.

Major issue (which must be addressed by the authors):

No additional experiments need to be done, but the authors should back off from their overly detailed model for the Rev/RRE component of the export complex because the information going into the modeling exercise is not sufficiently secure or precise to support the model. Issues limiting the precision of the reconstruction include: 1) The reconstruction is in negative stain and is at low resolution. 2) There seems to be a large amount of density missing from the complex. Even given that RNA does not stain well during negative staining, Rev-GB1 should stain well, and Rev could be seen as four round spheres in a previous reconstruction of Rev in complex with the RRE. The authors cite this paper, but do not compare that structure with the present structure, nor do they clearly state why there would be such a difference between the two (namely the complete lack of Rev-density in the current data). Furthermore, the Rev-RRE subcomplex adds up to about 75 kDa and should show up more strongly in the class averages than it does. This raises the concern that the suggested stoichiometry may be incorrect and/or that the averages acquired may include breakdown products and the preps used for EM may not have been exactly the same as those analyzed biochemically. 3) The (red) difference density that is shown in Figure 3, panels (a) and (b) is shallow and smeared a long way over the side of the complex and is hard to reconcile with the difference densities seen in the class averages shown in panel (c). 4) The interpretation of the difference density fails to mention the possibility that ordering of the Rev CTD may be taking place.

Similarly, the modeling has not (and probably cannot be) performed rigorously given the paucity of solid biochemical and structural information on the stoichiometry and orientation of the Rev-RRE complex. Issues include: 1) A lack of justification for the assumed jellyfish model and Rev stoichiometry in the context of the Crm1:RRE complex. 2) Questions surrounding the Rev subunits whose NES elements do not engage Crm1. The model suggests that six Rev monomers are correctly spaced to allow the terminal Rev monomers to interact with their binding sites on Crm1. However, it is not entirely clear that, based on the data in the accompanying crystal structure of a Rev dimer:RRE complex, the jellyfish model is the appropriate complex to consider. Furthermore, nothing is said of what happens with the intervening four monomeric tails. These would seem to be steric hindrances in the Crm1-Rev interaction, as it is not clear that the space between Rev and Crm-1 would allow nearly 23 kD of unstructured protein mass.

In essence, the reconstruction and the difference density reveal only the general site where the Rev-RRE complex binds the Crm1 dimer. This should be noted and is of some interest because it differs from the crystallographically-defined binding site for Snurportin1. However, we suggest backing off on the detailed interpretation of the complex structure (except for the Crm1 fitting, which appears solid) and omitting Figure 3 because there is no experimental evidence either here or in the companion paper for a Rev hexamer. Alternatively, the authors could consider obtaining a higher resolution cryo-EM reconstruction and/or providing much stronger biochemical evidence for the stoichiometry and structure of the bound Rev-RRE complex.

Other issues for the authors' consideration:

1) Abstract “Here we present the first view of the entire virus-host complex by single-particle electron microscopy.” Suggest rewording as “the assembled complex of known export factors”, explicitly include “negative stain” in the “single particle electron microscopy” (otherwise, people tend to assume that cryo-EM is meant) and to allow the possibility that (currently unknown) factors may be present in the functional complex.

2) Second paragraph of Introduction: “Here we report the single-particle electron microscopy (EM) structure of the Rev-RRE/Crm1-RanGTP export complex.” Caveat. While assembly of the complex is a notable achievement on its own, the wording here belies the reality that the difference density does not reasonably or unambiguously accommodate Rev or the RRE.

3) The text explanation of the experiment being performed in Figure 1 is incomplete/confusing. The experimental result is clear enough, but the authors should clarify the description of the source of the different samples and not force the reader to go through the experimental methods to understand what was done.

4) Figure 2 and Figure 2—figure supplement 2. A) Clarify that the reconstruction in Figure 2 is of the full Rev-RRE-Crm1-RanGTP complex. The figure legend needs to reflect this because otherwise it appears to be a reconstruction of a Crm1 dimer in the absence of Rev-RRE, which would contradict the main argument of the paper. B) It is difficult to believe the correlation coefficients of 0.97 and 0.98 for the claimed fits, which imply that the fits are essentially perfect. This is not the impression one gets from visual appraisal in which there are sizable unoccupied spaces inside the EM envelope. How were the volumes compared normalized? Are their densities sitting on top of large and equal constant densities? C) The fits shown in Figure 2—figure supplement 2 appear convincing, but the authors should provide statistical evidence that the crystallographic human Crm1 dimer structure does not fit well into the mCrm1/Ran(GTP) complex and that the separated monomers from the fit of the mCrm1/Ran(GTP) cannot be well accommodated by the hCrm1 dimer density.

5) “…by single-particle negative-stain EM using random conical tilt…”: Give some brief general account of how random conical tilt works and why it is useful, and give a couple of references. The vast majority of potential readers will never have heard of RCT and the Methods section is not much help.

Similarly, the authors should add a sentence or two that helps the general reader to understand that Ran(GTP) binds on the interior of the Crm1 solenoid and stabilizes a strained conformation that creates the hydrophobic NES binding cleft (and also comment on whether the Ran(GTP) complex is required to explain the observed EM density).

6) The authors state: “Rev and the RRE are unambiguously positioned on one side of the Crm1 dimer, with Rev close to Crm1 and the RRE slightly further away” Where is the evidence for the latter part of that sentence?

7) A crystal structure of the RanGTP-CRM1 complex bound to the Rev NES has been reported (28). Why hasn't the footprint of this Rev NES-Crm1 complex been compared with the data presented herein?

8) An example of the overly detailed modeling/interpretation occurs in the Results section: “Given the flexible nature of the Rev carboxy-terminus, there may be some variation in this distance that allows an NES from a neighboring subunit to be utilized, potentially increasing avidity from a high local concentration of NESs”. The authors' idea seems to be that avidity effects are (relatively) low because the NESs from the terminal Rev molecules in the hexamer are normally used and that avidity effects would be increased if NESs from non-terminal Rev molecules were used. This is an overly detailed and relatively weak argument: why not just leave in the final two sentences, which seem to cover how RNA-dependent remodeling of the REV assembly could modulate Crm1 binding affinity?

9) The authors should elaborate on the meaning of the observed Hill coefficients for murine and human Crm1 binding to the Rev-RRE complex (2.12 and 2.37, respectively). Unless we're missing something, the Hill coefficient should not actually exceed 2 in either case (i.e., the number of Crm1 molecules that bind the Rev-RRE complex). Also, why do the authors think that the mCrm1 molecules bind cooperatively to the Rev-RRE complex if they don't actually touch one another? Finally, it is difficult to believe that the Hill coefficients of 2.37 +/- 0.13 and 2.12 +/- 0.25 are really different at the 0.0001 confidence level. Please double check this.

10) It is a bit surprising that mCrm1 does not dimerize if this oligomeric state is used in the export of some human RNA complexes, and this issue probably merits comment.

---

## [Author Response]

*Major issue (which must be addressed by the authors)*:

*No additional experiments need to be done, but the authors should back off from their overly detailed model for the Rev/RRE component of the export complex because the information going into the modeling exercise is not sufficiently secure or precise to support the model. Issues limiting the precision of the reconstruction include: 1) The reconstruction is in negative stain and is at low resolution. 2) There seems to be a large amount of density missing from the complex. Even given that RNA does not stain well during negative staining, Rev-GB1 should stain well, and Rev could be seen as four round spheres in a previous reconstruction of Rev in complex with the RRE. The authors cite this paper, but do not compare that structure with the present structure, nor do they clearly state why there would be such a difference between the two (namely the complete lack of Rev-density in the current data)*.

We thank the reviewers for pointing this out. Indeed, the extra density in the complex is not large enough to position Rev-RRE for a number of reasons. Perhaps the most obvious is that RNA does not stain as well as protein. The previously published class averages of Rev-RRE alone (11) suffer from the same problem as the current study of only having a very weak or no RNA density, and the dimensions of the particles ∼80-100 Å are consistent with those of the Rev oligomer without visible RNA.

Crm1 accounts for about two-thirds of the mass of the whole particle and 80% of the total protein mass so it is not surprising that Crm1 dominates the alignment. The Rev-RRE density is ‘averaged-out’ because it binds heterogeneously. Additionally, because images are two-dimensional projections of a three-dimensional object, some of the Rev-RRE density may lie on top of the Crm1 dimer and be partially occluded in class averages. Better density was observed for Rev-RRE in our previous publication because it was the only particle being aligned.

Furthermore, to gain confidence in the assignment of density to Rev-RRE, we prepared samples with Rev fused to GB1, which has a ∼8 kD mass that sums to a total of 48 kD in the complex, to add extra density to visualize directly the location of Rev-RRE in 2D class averages. Indeed, the extra density of GB1 observed confirmed our assignment of Rev-RRE in the reconstruction.

*Furthermore, the Rev-RRE subcomplex adds up to about 75 kDa and should show up more strongly in the class averages than it does. This raises the concern that the suggested stoichiometry may be incorrect and/or that the averages acquired may include breakdown products and the preps used for EM may not have been exactly the same as those analyzed biochemically*.

While in principle the Rev oligomer should provide ∼75 kD of mass, it is known from NMR experiments that almost half of the protein is unstructured and unlikely to be visible in the class averages. This unstructured portion is mostly in the C-terminus and not expected to become ordered in the complex (see point 4 below). Of the structurally defined portion, the oligomerization domains form the core of the structure and would be the most visible at a resolution of 25 Å, but the arginine rich motifs, which extend away from the core, are ∼20 Å wide and have ∼30 Å gaps between adjacent subunits so may be less visible. So the observed EM density represents approximately the expected mass of the core oligomer given the flexibility in the interaction between Crm1 and Rev-RRE. Our assignment of Rev-RRE in the complex is supported by tagging Rev with GB1, tagging the RRE with streptavidin, and also by observing class averages with electron cryo-microscopy (Figure 3), which shows better density of RNA. All support the positioning of the complex and begin to show the position of the RNA.

There is some concern about complex dissociation, and indeed we observe a minor population of free Crm1 monomers. We, however, never see Crm1 dimers in the absence of Rev-RRE and the dimerization affinity is very weak so the single particles used for the reconstruction are almost certainly full complexes. Given the cooperative nature of the Rev-RRE interaction, we also do not expect partial Rev complexes to be stable, although we cannot rule out the possibility. The expected conformational heterogeneity of the docked Rev-RRE complex and the inability to see the RNA by negative stain are expected to be the major factors in reducing the observed density.

*3) The (red) difference density that is shown in*
Figure 3*, panels (a) and (b) is shallow and smeared a long way over the side of the complex and is hard to reconcile with the difference densities seen in the class averages shown in panel (c)*.

We show in Figure 3—figure supplement 1 that increasing the number of particles for sub-classification partially reveals density where Rev-RRE is localized. The extra densities, however, appear in a variety of orientations relative to the Crm1 dimer and indicate flexibility or many different binding orientations between Rev-RRE and Crm1. Combining particles from many different class averages is expected to smear the density and cause the average to be weaker.

The GB1-Rev tagged class average also displays a ‘smear’ of additional density consistent with flexibility in the docking orientations. The extra density observed extends beyond the envelope of the reconstruction, as expected for the additional appended mass. The streptavidin-labeled RRE sample adds 60 kD of mass, also outside the envelope, and the cryo-EM averages bring out features of the RNA not visible by negative stain. We have since increased the number of cryo particles, which reveal some preferred orientations of binding to the grids and does not allow calculating a reliable reconstruction.

*4) The interpretation of the difference density fails to mention the possibility that ordering of the Rev CTD may be taking place*.

While possible, the crystal structure of the Rev NES bound to Crm1 and supporting NMR experiments indicate ordering of the C-terminus is probably limited to the portion in direct contact with Crm1 because residues on either side of the NES maintain a high degree of flexibility (28). Heteronuclear NOE experiments previously showed that the C-terminus is flexible, albeit not in the complex (11).

*Similarly, the modeling has not (and probably cannot be) performed rigorously given the paucity of solid biochemical and structural information on the stoichiometry and orientation of the Rev-RRE complex. Issues include: 1) A lack of justification for the assumed jellyfish model and Rev stoichiometry in the context of the Crm1:RRE complex. 2) Questions surrounding the Rev subunits whose NES elements do not engage Crm1. The model suggests that six Rev monomers are correctly spaced to allow the terminal Rev monomers to interact with their binding sites on Crm1. However, it is not entirely clear that, based on the data in the accompanying crystal structure of a Rev dimer:RRE complex, the jellyfish model is the appropriate complex to consider. Furthermore, nothing is said of what happens with the intervening four monomeric tails. These would seem to be steric hindrances in the Crm1-Rev interaction, as it is not clear that the space between Rev and Crm-1 would allow nearly 23 kD of unstructured protein mass*.

*In essence, the reconstruction and the difference density reveal only the general site where the Rev-RRE complex binds the Crm1 dimer. This should be noted and is of some interest because it differs from the crystallographically-defined binding site for Snurportin1. However, we suggest backing off on the detailed interpretation of the complex structure (except for the Crm1 fitting, which appears solid) and omitting*
Figure 3
*because there is no experimental evidence either here or in the companion paper for a Rev hexamer. Alternatively, the authors could consider obtaining a higher resolution cryo-EM reconstruction and/or providing much stronger biochemical evidence for the stoichiometry and structure of the bound Rev-RRE complex*.

We believe there is ample evidence to support a Rev hexamer in our complex. Previous work accurately determined the stoichiometry of Rev bound to the 245nt RRE used here (11), and we have further corroborated the stoichiometry of Rev-RRE by measuring the mass of the Rev-RRE complex as well as the masses of Crm1, Ran^GTP^, and the complex formed with all of the components using multi-angle laser light scattering. Those data are presented in the new Figure 1—figure supplement 1, along with an additional measurement of the stoichiometry using protein standards from amino acid analysis to quantify the concentrations of each component after purification through size-exclusion chromatography.

The jellyfish model cannot be fully evaluated in the context of the current EM data because we do not know the details of how the RRE fits into the structure, however the broader point that the NESs extend away from the Rev-RRE complex to engage Crm1 remains the simplest supposition and is entirely consistent with the EM data. We can reasonably position the Rev oligomer with respect to the Crm1 dimer because the Rev NES bound to Crm1 has a defined orientation that points the amino-terminal end of the NES in the direction of the C-terminal of the Rev oligomer to which NESs connect through a 12 amino acid linker, extending as much as ∼40 Å. Steric clashes are likely mitigated by the inherent flexibility of this portion of Rev.

Although we can infer these aspects of the overall orientation of components, we agree that the uncertainties of positioning Rev-RRE into the density do not warrant detailed models for how the hexamer fits into the Crm1 dimer complex. We have removed the relevant discussion points as well as previous Figure 3.

We wish to emphasize that we do not believe that Rev-RRE docks to the Crm1 dimer in a single way, and indeed, our binding analyses are consistent with an avidity argument for binding to multiple NESs, as described below. Nonetheless, the accompanying crystal structure paper provides some structural rationale to reconcile the stoichiometry of a Rev hexamer docking to a Crm1 dimer.

*Other issues for the authors' consideration*:

*1) Abstract “Here we present the first view of the entire virus-host complex by single-particle electron microscopy.” Suggest rewording as “the assembled complex of known export factors”, explicitly include “negative stain” in the “single particle electron microscopy” (otherwise, people tend to assume that cryo-EM is meant) and to allow the possibility that (currently unknown) factors may be present in the functional complex*.

We changed the sentence to indicate that this represents “a view of an assembled HIV-host nuclear export complex”. Our data also includes class averages from cryo EM to support the negative stain data.

*2) Second paragraph of Introduction: “Here we report the single-particle electron microscopy (EM) structure of the Rev-RRE/Crm1-RanGTP export complex.” Caveat. While assembly of the complex is a notable achievement on its own, the wording here belies the reality that the difference density does not reasonably or unambiguously accommodate Rev or the RRE*.

We now refer to the overall architecture of the assembly and do not describe it as the structure.

*3) The text explanation of the experiment being performed in*
Figure 1
*is incomplete/confusing. The experimental result is clear enough, but the authors should clarify the description of the source of the different samples and not force the reader to go through the experimental methods to understand what was done*.

We have corrected the error.

*4)*
Figure 2
*and*
Figure 2—figure supplement 2*. A) Clarify that the reconstruction in*
Figure 2
*is of the full Rev-RRE-Crm1-RanGTP complex. The figure legend needs to reflect this because otherwise it appears to be a reconstruction of a Crm1 dimer in the absence of Rev-RRE, which would contradict the main argument of the paper*.

We have revised as suggested.

*B) It is difficult to believe the correlation coefficients of 0.97 and 0.98 for the claimed fits, which imply that the fits are essentially perfect. This is not the impression one gets from visual appraisal in which there are sizable unoccupied spaces inside the EM envelope. How were the volumes compared normalized? Are their densities sitting on top of large and equal constant densities*?

We revised the fitting procedure as described in the Methods and recalculated the correlation coefficients presented in Figure 2—figure supplement 2.

*C) The fits shown in*
Figure 2—figure supplement 2
*appear convincing, but the authors should provide statistical evidence that the crystallographic human Crm1 dimer structure does not fit well into the mCrm1/Ran(GTP) complex and that the separated monomers from the fit of the mCrm1/Ran(GTP) cannot be well accommodated by the hCrm1 dimer density*.

We present the correlation coefficients as a measure of the fit, but given the low resolution we cannot use this parameter alone to determine the best fit. We use mutagenesis and biochemical data in conjunction with EM to assess the quality of fit, as described.

We do not have a reconstruction of Rev-RRE bound mCrm1 and Ran^GTP^ into which we can compare the fit of the human Crm1 dimer. We have included gel filtration profiles and class averages that show qualitative differences between the assembly and shape of complexes formed with either murine or human crm1 (Figure 2).

*5) “…by single-particle negative-stain EM using random conical tilt…”: Give some brief general account of how random conical tilt works and why it is useful, and give a couple of references. The vast majority of potential readers will never have heard of RCT and the Methods section is not much help*.

We have added text to address this point.

*Similarly, the authors should add a sentence or two that helps the general reader to understand that Ran(GTP) binds on the interior of the Crm1 solenoid and stabilizes a strained conformation that creates the hydrophobic NES binding cleft (and also comment on whether the Ran(GTP) complex is required to explain the observed EM density)*.

We now include text on the need for Ran^GTP^ and its function.

*6) The authors state: “Rev and the RRE are unambiguously positioned on one side of the Crm1 dimer, with Rev close to Crm1 and the RRE slightly further away” Where is the evidence for the latter part of that sentence*?

This conclusion came from comparing the location of extra densities from tagged versions of Rev or the RRE, but given the caveats above, we have eliminated the latter half of the sentence.

*7) A crystal structure of the RanGTP-CRM1 complex bound to the Rev NES has been reported (*[28]*). Why hasn't the footprint of this Rev NES-Crm1 complex been compared with the data presented herein*?

This structure is represented in Figure 2—figure supplement 2 with the footprint of the NES binding site indicated as the cyan-colored residues and labeled ‘NES site’. This same structure is shown in Figure 3 with the Rev NES bound to it. In other models presented, the NES site has the same coloring because the Rev NES establishes essentially the same contacts as the Snurportin-1 NES (28).

*8) An example of the overly detailed modeling/interpretation occurs in the Results section: “Given the flexible nature of the Rev carboxy-terminus, there may be some variation in this distance that allows an NES from a neighboring subunit to be utilized, potentially increasing avidity from a high local concentration of NESs”. The authors' idea seems to be that avidity effects are (relatively) low because the NESs from the terminal Rev molecules in the hexamer are normally used and that avidity effects would be increased if NESs from non-terminal Rev molecules were used. This is an overly detailed and relatively weak argument: why not just leave in the final two sentences, which seem to cover how RNA-dependent remodeling of the REV assembly could modulate Crm1 binding affinity*?

We did not mean to imply that there is necessarily one discrete binding mode but rather believe that avidity plays a significant role in complex assembly, using multiple binding modes that engage different flexible NESs and possibly different forms of the Rev oligomer. We have limited this discussion and refer to the RNA-dependent remodeling shown in the companion crystal structure paper.

*9) The authors should elaborate on the meaning of the observed Hill coefficients for murine and human Crm1 binding to the Rev-RRE complex (2.12 and 2.37, respectively). Unless we're missing something, the Hill coefficient should not actually exceed 2 in either case (i.e., the number of Crm1 molecules that bind the Rev-RRE complex). Also, why do the authors think that the mCrm1 molecules bind cooperatively to the Rev-RRE complex if they don't actually touch one another*?

Prompted by this comment, we have re-evaluated fitting the Hill model to the binding data and used an alternative binding model (in the text and Figure 2). The McGhee-von Hippel model may more realistically represent the binding mode of Crm1 to the Rev-RRE complex having an array of NESs. Both models yield the same overall conclusion that human Crm1 results in a more cooperative complex, presumably due to its dimeric interface. The mouse Crm1 subunits likely also touch each other but more weakly than the human dimer, and are stabilized when binding Rev-RRE.

*Finally, it is difficult to believe that the Hill coefficients of 2.37 +/- 0.13 and 2.12 +/- 0.25 are really different at the 0.0001 confidence level. Please double check this*.

We have corrected these errors.

*10) It is a bit surprising that mCrm1 does not dimerize if this oligomeric state is used in the export of some human RNA complexes, and this issue probably merits comment*.

Sherer et al have examined the evolutionary differences and we have expanded the discussion and extended some of their initial observations.